# Microbial Biofilms at Meat-Processing Plant as Possible Places of Bacteria Survival

**DOI:** 10.3390/microorganisms10081583

**Published:** 2022-08-06

**Authors:** Yury Nikolaev, Yulia Yushina, Andrey Mardanov, Evgeniy Gruzdev, Ekaterina Tikhonova, Galina El-Registan, Aleksey Beletskiy, Anastasia Semenova, Elena Zaiko, Dagmara Bataeva, Ekaterina Polishchuk

**Affiliations:** 1Federal Research Center “Fundamentals of Biotechnology” of RAS, Leninsky Prospect, 14, 119991 Moscow, Russia; 2V.M. Gorbatov Federal Research Center for Food Systems of RAS, Talalikhina St., 26, 109316 Moscow, Russia

**Keywords:** microbial biofilm, meat-processing, foodborne pathogens

## Abstract

Biofilm contamination in food production threatens food quality and safety, and causes bacterial infections. Study of food biofilms (BF) is of great importance. The taxonomic composition and structural organization of five foods BF taken in different workshops of a meat-processing plant (Moscow, RF) were studied. Samples were taken from the surface of technological equipment and premises. Metagenomic analysis showed both similarities in the presented microorganisms dominating in different samples, and unique families prevailing on certain objects were noted. The bacteria found belonged to 11 phyla (no archaea). The dominant ones were *Actinobacteria*, *Bacteroidetes*, *Firmicutes*, and *Proteobacteria*. The greatest diversity was in BFs taken from the cutting table of raw material. Biofilms’ bacteria may be the cause of meat, fish and dairy products spoilage possible representatives include *Pseudomonas*, *Flavobacterium*, *Arcobacter*, *Vagococcus*, *Chryseobacterium*, *Carnobacterium*, etc.). Opportunistic human and animal pathogens (possible representatives include *Arcobacter*, *Corynebacterium*, *Kocuria*, etc.) were also found. Electron-microscopic studies of BF thin sections revealed the following: (1) the diversity of cell morphotypes specific to multispecies BFs; (2) morphological similarity of cells in BFs from different samples, micro-colonial growth; (3) age heterogeneity of cells within the same microcolony (vegetative and autolyzed cells, resting forms); (4) heterogeneity of the polymer matrix chemical nature according to ruthenium red staining.

## 1. Introduction

The main form microorganisms in natural and anthropogenic systems are biofilm. Biofilms (BFs) are monospecies or multispecies microbial communities enclosed in a self-produced polymer matrix attached to biotic or abiotic surfaces [1]. Biofilms, as self-organized communities of bacteria developing as a tolerant phenotype in polymer matrix and attached to abiogenic or biogenic surfaces, are a serious threat to the food industry [2,3]. Extensive information is available on the significantly greater resistance of microorganisms in BFs to damaging influences, including disinfectants, compared to planktonic cultures [4]. Therefore, biofilm contamination of food production facilities and methods of effective BF removal from production surfaces and equipment are a huge problem [5,6]. Biofilms are responsible not only for equipment damage and food spoilage, but also for survival and spread of pathogenic bacteria [3,4] and causes diseases related to foodborne pathogens. Thus, according to monitoring studies, up to 80% of bacterial infections in the United States are directly associated with foodborne pathogens [7]. In addition, food raw materials entering the enterprises may be contaminated by resting forms of various (not only foodborne) pathogens that survive in this state in natural ecotopes when the host changes. In addition, biofilms are considered as a source of cross-contamination in food processing plants [7,8].

Despite the extensive BF studies in the food industry [5,6,8] a number of problems have not been solved, while others require refinement and clarification, in particular, autoregulation of BF development; mechanisms of BF resistance to damaging influences. Furthermore, we need to improve detection efficiency for BFs and seasonal dynamics of their formation and to develop effective methods and tools to control biofilm formation, particularly at food production facilities. The issues of BF resistance to toxicants (antibiotics, disinfectants), physical factors (UV, etc.) and the role of biofilm growth in the development of antibiotic resistance (AR) are especially relevant, not only due to greater horizontal transfer of AR determinants in BFs [2], but also due to the frequency of AR mutants occurrence [9,10].

Combination of modern methods of molecular-genetic and microbiological analyzes and increasing amount of information on the importance of intercellular interactions in complex communities has led to extensive studies on interactions between different bacterial species in multispecies biofilm consortia [11]. At the same time, along with the recognition of interspecific relations role in the stability of multispecies BFs [5,12], an understanding is being formed of possibly even greater significance of age heterogeneity in the biofilm population (sensitive dividing cells and resting stress-resistant forms), as well as intrapopulation phenotypic heterogeneity (persister cells, small-colony variants (SCV)) [13,14]. Although multispecies BFs are predominant form of food contamination, most studies on the properties and mechanisms of biofilm formation have been carried out on model monospecies BFs [15]. Therefore, the study of the taxonomic composition and structural organization of native food biofilms is of great practical and theoretical interest.

Unfortunately, it should be noted that insufficient attention is paid to studying the biofilm contamination at food production facilities in Russia. At the same time, the development of effective methods and tools for BF control in food industry is possible only on the basis of versatile studies of biofilm formation, taking into account regional conditions, local specifics of incoming raw materials, conditions of its processing, etc.

The aim of this study was to detect biofilms in various technological zones of a meat processing plant in Moscow, Russia as a possible source of contaminating microflora and to determine biofilm consortium taxonomic composition, structural organization, morphological and age heterogeneity.

This study is the basis for further research of the biofilm formation seasonal dynamics at meat processing plants, changes in the taxonomic, age and structural characteristics of BFs, in order to develop effective methods to suppress the development and spread of biofilm microorganisms, including pathogens, at food processing plants.

## 2. Materials and Methods

### 2.1. Research Subjects

The work investigated biofilms collected from the surfaces of technological equipment in various zones at a meat processing plant (MPP) in the central region of Russia.

Samples were taken before the routine disinfection procedure (daily treatment of all technological surfaces with sodium hypochlorite solution) from technological equipment or premise surfaces (Table 1) by scraping with a metal spatula. This time point was chosen to catch both young and mature BFs. In preliminary experiments, we detected BF in all sampling points both before and after disinfection. Detection of BF was made by the use of two BF features: slime formation (detected by fingers), and bubble formation out of H_2_O_2_ (3%). The biological material was immediately placed in sterile saline (for taxonomic studies) or in a solution of glutaraldehyde in cacodylate buffer (for electron-microscopic studies).

### 2.2. Microscopic Examinations

The ultrastructural organization of biofilm samples was studied using transmission electron microscopy (TEM) of their ultrathin sections.

Biofilm samples taken for electron microscopy were fixed with a solution of 2.5% glutaraldehyde in cacodylate buffer (0.05 M sodium cacodylate solution, pH 7.0–7.5) and stored at 4 °C for one day; then washed three times with the same buffer solution for 5 min and fixed in a solution of OsO_4_ (1% OsO_4_)-0.7% solution of ruthenium red (Sigma, St. Louis, MO, USA) in cacodylate buffer) at 4 °C for 1.5 h. After fixation, the samples were placed in 2% agar-agar and incubated in 3% solution of uranyl acetate in 30% ethyl alcohol for 4 h and then in 70% ethanol for 12 h at 4 °C. The material was dehydrated in 96% ethanol (2 times for 15 min) and then in absolute acetone (3 times for 10 min). The samples were impregnated with EPON-812 resin (EpoxyEmbedding Medium Epon^®^ 812, Sigma-Aldrich, St. Louis, MO, USA) and stored in resin:acetone mixture with a ratio of 1:1 for 1 h and then in resin:acetone mixture with a ratio of 2:1 for 1 h. The resulting material was filled in capsules with resin and polymerized at 37 °C for one day and then at 60 °C for another day. Ultrathin sections were obtained using LKB-III microtome (LKB, Stockholm, Sweden) and contrasted in an aqueous solution of 3% uranyl acetate (for 30 min) and then in an aqueous solution of 4% lead citrate (for 30 min).

To identify acidic mucopolysaccharides in biofilms, ruthenium red stain (Sigma, St. Louis, MO, USA) was used. It was added in an amount of 0.7% together with OsO_4_, with which it interacted. The presence of extracellular polysaccharides in biofilms of various bacteria was shown using ruthenium red (Sigma, St. Louis, MO, USA) [16].

The obtained slides were examined visually using JEM 100SHP electron microscope (JEOL, Akishima, Japan) at accelerating voltage of 80 kV and operating magnification of 5000–50,000. The materials were photographed using Morada G2 digital image output system (EMSIS GmbH, Münster, Germany).

Identification of physiological type of cells was performed by the similarity method: i.e., cells of typical structure were considered as vegetative ones, cells with thick cell wall and/or dark cytoplasm were considered as resting forms [17], empty envelopes were considered as lysed cells, “hairy” cells were considered as persisters [18].

### 2.3. Taxonomic Analysis of Biofilms

Metagenomic DNA from biofilm samples was isolated using DNeasy PowerSoil Kit (Qiagen, Hilden, Germany) according to the manufacturer’s protocols. Two sets of primers were used to amplify the V3–V4 variable region of the 16S rRNA gene: universal 341F CCTAYGGGDBGCWSCAG and 806R GGACTACNVGGGTHTCTAAT [19]. PCR fragments were sequenced using Illumina MiSeq (Illumina, San Diego, CA, USA). Readings from all samples were combined. Low quality readings, singletons and chimeras were excluded. The remaining readings were clustered into operational taxonomic units (OTUs) with a minimum identity of 97%. To determine the proportion of OTUs in each of the samples, the original readings (including low-quality ones and singletons) with a minimum identity of 97% along the entire reading were overlaid on the representative OTU sequences. Usearch 11 [20]. was used to perform all of these procedures. Taxonomic identification of microorganisms by 16S rRNA gene sequences was performed using Usearch and Silva v. 1.2.11 database with default parameters [21,22]. For each sample, at least 5000 thousand sequences were obtained. Differences in the diversity among the samples were calculated using Shannon, Simpson and Chao1. Chao1, based on Shannon, and Observed-species indices, rarefactions curves were prepared via QIIME (version 1.7.0).

## 3. Results

### 3.1. Microscopic Examination of Biofilms

Microscopy has a great potential for studying biofilms, including the methods of transmission electron microscopy (TEM) used in this work to study ultrathin sections in combination with the cytochemical analysis of acidic mucopolysaccharides of the biofilm matrix.

Biofilms sampled in different places and workshops of MPP differed in thickness and consistency (Table 1). Sample 3, in contrast with others, was a thin dry biofilm (BF) unsuitable for obtaining and studying ultrathin sections. It has been studied only by metagenomic analysis methods. Microscopic examination of ultrathin BF sections revealed a good development of the biopolymer matrix in all samples, which differed among samples in density and chemical composition. In samples 2, 4 and 5, along with the polysaccharide matrix stained with ruthenium red, unstained pericellular regions of the matrix were found, apparently of a protein or nucleic nature. All samples were represented by multispecies biofilms, heterogeneous both in bacterial cell morphology and their assumed physiological age.

Sample 1 from a cutting table in the deboning and trimming workshop had the greatest heterogeneity of the microbial population in terms of cell morphology and their physiological age (Figure 1 and Figure 2). Very large (more than 2 µm) meat cells captured in BF and bacterial cells of several morphotypes are visible (Figure 1). At higher magnification (Figure 2), membrane vesicles (V) are found on cells and also in the matrix.

It should be noted that unusual “hairy” cells are found both at low (Figure 1) and high (Figure 2) magnification.

BF #1 matrix, similar to the rest of the samples, was formed by polysaccharides and stained well with ruthenium red.

Sample 2 was also a multispecies biofilm. Compared to sample 1, most of the bacteria of the consortium in it were represented by gram-negative vegetative cells (Figure 3), which is clearly seen at high magnification (Figure 4). Along with vegetative cells, BF contained resting forms (RF) with a thickened cell membrane, electron-dense cytoplasm, and a compacted nucleoid (CN) located in the cell center. A typical characteristic for the biofilms of this sample, as well as samples 4 and 5, was a well-formed, but chemically heterogeneous matrix. The main volume of BF was represented by a polysaccharide matrix with heterogeneous staining by ruthenium red. In the stained matrix, its inhomogeneous fibrous structure (FS) was seen, clearly visible at higher magnification (Figure 4) and found in BFs of other samples (1, 4, 5). Perhaps this structure contributes to a stronger interaction of cells. Along with a dense polysaccharide matrix, it contained large areas poorly stained or not stained with ruthenium red. They formed as a capsular layer (CL) around single cells and cell groups (Figure 3 and Figure 4) and apparently contained non-carbohydrate polymers (proteins, nucleic acids). This matrix heterogeneity apparently reflects the taxonomic heterogeneity of biofilms but does not depend on the physiological age of cells, since vegetative cells (VC) and RF are detected in the same locus (Figure 4).

The pronounced age heterogeneity of the cells in biofilm consortium was found, with various cell types simultaneously present in the microzone(s): vegetative cells; resting forms characterized by a thickened cell wall, electron-dense cytoplasm (compared to vegetative cells), and compacted nucleoid (CN); as well as autolyzed cells (A) with only residual cell membrane.

Sample 4 was a multispecies biofilm with 5 types of morphologically different cells, many of which were similar to cells from samples 1 and 2 (Figure 5). On the sections, vegetative and resting cells are well identified; as well as cells with multiple inclusions, apparently of polyoxybutyric acid (POBA), which accumulates in stationary and resting cells.

The formation of non-polysaccharide capsular layer around the cells or cell groups was also noted. Figure 5 shows various morphological types of cells. Their microcolonial growth was well demonstrated by the example of type II cells (rod-shaped) visible in the micrograph as elongated cells and their compactly packed cross sections (VC II) (Figure 5).

Sample 5 was also represented by multispecies biofilm, but with the following structural characteristics in the ultrathin sections of this BF. First, zones of two types were found in the polymer matrix. In one zone, the matrix was organized in the form of dense veiny structures (Figure 6A), which were unevenly stained with ruthenium red. The matrix contained cells of a gram-positive bacterium (by the structure of the cell membrane) with microcolonial growth (Figure 6A,B). Within the same microcolony, cells of different physiological ages were detected, i.e., vegetative cells (VC) dividing by septation (having pronounced division septa, DS) and thick-walled resting forms (RF) (Figure 6B). A capsular layer was formed around the cells, which was not stained with ruthenium red. It should be noted that in sample 2 (Figure 4) the capsular layer was formed by gram-negative bacteria, while in sample 5 it was formed by gram-positive bacteria.

### 3.2. Taxonomic Analysis of Biofilms

Biofilm taxonomic composition of five samples was determined based on the analysis of the variable V3–V4 region of the 16S rRNA gene. After processing the obtained data, 86,017 sequences were obtained for all samples, which were combined into a cluster of 470 operational taxonomic units with a minimum identity of 0.97. This allows complete description of biodiversity in the studied communities, which is confirmed by the calculated indices of species diversity (Table 2).

In particular, the highest Chao1 value obtained for sample 1 was 325.5; in other samples it was lower. Thus, the obtained data are sufficient for complete description of biodiversity in the studied samples.

Taxonomic analysis showed that all OTUs belonged to bacteria. Archaea were not found in the studied communities. All found OTUs belong to 11 phyla (Figure 7). In all samples, the dominant microorganisms were representatives of the following phyla: *Actinobacteria*, *Bacteroidetes*, *Firmicutes*, *Proteobacteria*.

In three samples (2, 3, 5), representatives of the *Bacteroidetes* phylum dominated, which accounted for more than half of the data obtained. In sample 1, *Bacteroidetes* (38.28%) were inferior to *Proteobacteria* (42.83%). The most uniform distribution of shares between phyla was observed in sample 4: *Actinobacteria*-36.29%, *Firmicutes*-32.83%; *Bacteroidetes*-22.27%. Minor groups included representatives of *Acidobacteria*, *Chloroflexi*, *Cyanobacteria*, *Epsilonbacteraeota*, *Fusobacteria*, *Patescibacteria*, *Planctomycetes*, *Verrucomicrobia*, etc.

#### 3.2.1. Identification of the Exclusive and Basic Microbiota in the Food Industry

Our analysis revealed a total of 101 families. Of the 101 identified families, 18 (15.5%) were common to all biofilms, and 83 (84.7%) were specific (Appendix A).

Biofilm from the joint of the wall and the cutting table (sample 1) was characterized by the highest diversity of microbial community. The *Flavobacteriaceae* family was represented in all samples, and in some their number reached more than 50% (in sample 2—55.09%, in sample 3—54.30%) (Figure 8). However, in sample 4 it was presented in the smallest amount compared to other samples (0.75%).

One of the most common families were *Pseudomonadaceae* and *Moraxellaceae*. Representatives of the *Pseudomonas* were found in all samples. The largest number of them were identified in sample 1 (18.64%), followed by sample 4 (2.43%), and the smallest number of representatives of the *Pseudomonas* were found in samples 2, 3 and 5 (1.37%, 1.58% and 0.12%, respectively).

*Crocinitomicaceae*, which includes representatives of the genus *Fluviicola*, was found in all five biofilms. The largest number of them was found in sample 2, which was taken from the drain ladder of the cutting shop and amounted to 7.88%.

Representatives of the *Enterococcaceae* are known for their ability to form biofilms, were found at all points of the study, the largest number of them was found in sample 4 and amounted to 12.47%, and the smallest number of them in sample 1 (0.13%).

Samples 2, 3 and 5 were represented by biofilms, which were quite similar in taxonomic composition: the dominant OTUs are *Flavobacteriaceae*. Another dominant family in sample 3 belongs to the *Burkholderiaceae* of the *Proteobacteria*.

Sample 4 was represented by the most distinctive microbial community. First, in this community, the species diversity was the lowest, with only 125 OTUs. Second, the dominant microorganisms were the representatives of the following phyla: *Actinobacteria-Corynebacteriaceae* (16.29%), *Actinobacteria-Micrococcaceae* (15.04%), *Firmicutes-Enterococcaceae* (12.47%), *Bacteroidetes-Sphingobacteriaceae* (11.11%), *Bacteroidetes-Weeksellaceae* (10.36%). There are human pathogens among bacteria of the *Corynebacterium* genus. Another eurybionts dominating in sample 4 community are *Sphingobacteria*.

In sample 5, in addition to *Flavobacteriaceae*, the dominant components of the community were representatives of the *Enterococcaceae* (8.46%) and *Arcobacteraceae* (6.06%). Furthermore, *Enterococcaceae* were presented in samples 2 (4.90%), 3 (1.09%) and 4 (12.47%), and in a minimum amount in sample 1 (0.13%).

At some points, a unique microflora was noted. Thus, representatives of the families *Aerococcaceae* and *Carnobacteriaceae* were found at all sampling points except for point 1 (Appendix A). Representatives of the *Clostridiaceae* were found in samples 2, 4 and 5 (Appendix A). The largest number of them was presented in sample 5 and amounted to 1.96%. Representatives of the family *Lactobacillaceae*, *Listeriaceae* and *Staphylococcaceae* (Appendix A) were found mainly only in sample 4, which was selected in the forming workshop at the corner of the vacuum filler table.

Thus, the microbial communities of samples 1–5 contained microorganism’s characteristic of food biofilms, including those involved in food spoilage. The microbial communities of the analyzed samples differed in their taxonomic composition and had high diversity. They included representatives of 11 phyla, among which several dominant families may be distinguished. A characteristic of microbial communities in samples 1, 2, 3, 5 was the presence of the *Flavobacteriaceae* representatives among the key dominants.

According to our results, pathogenic bacteria associated with foodborne diseases, such as *L. moncytogenes*, *Salmonella* spp., *S. aureus* and *Campylobacter* spp., were not identified in our samples.

#### 3.2.2. Prevalence of Representatives of the *Enterobacteriaceae* Family

Representatives of the *Enterobacteria* were identified in small numbers but were found in all samples. The largest number of them were found in sample 2 and were represented by the genera *Serratia* and *Morganella*. The next in the predominance of representatives of the Enterobacteria family was sample 3 and 5 and were represented by the genera *Budvicia* and *Morganella* in different quantities. In the remaining samples, representatives of the Enterobacteria family were found in insignificant quantities.

## 4. Discussion

It should be noted that despite strict compliance with sanitization and disinfection measure at the studied MP, BFs were detected as early as 10–24 h after sanitization. This confirms the available information on the rapid (several hours) BF formation by bacteria, both remaining viable after sanitization and entered from external environment [23]. Our data are in agreement with this. We observed both young and old BF (as indicated by polyoxybutyric acid and resting forms presence for mature BFs and by vegetative cells for young BFs).

In this work food biofilms were sampled in different places (on cutting tables, forming equipment, and drainpipe walls), metagenomic analysis showed a high taxonomic diversity of bacteria dominating in different samples (Appendix A). According to the results of 16S rRNA analysis, representatives of Actinobacterium, Bacteroidetes, Firmicutes, and Proteobacteria dominated in all biofilm samples. By the indices of species diversity, the highest taxonomic diversity was in sample 1 taken at the drain of the cutting table in the deboning and trimming workshop, where incoming meat raw materials are processed. Taxonomic analysis results of the sampled BFs (Figure 7) were confirmed by the microscopic studies of BF thin sections, which revealed the morphological similarity of cells in BF samples.

The studied characteristics of biofilm contamination allow concluding that the dominant microflora in different technological zones of a meat processing enterprise is sufficiently similar. Currently, extensive evidence has been obtained indicating a significantly higher resistance of multispecies biofilms to damaging physical influences, chemical and biological toxicants, compared to monospecies biofilms [24]. A number of studies have described the taxonomic composition of pathogens forming food BFs, among which the most common species are: *Escherichia coli* [25]; *Staphylococcus aureus* [26]; *Listeria monocytogenes* [27]; *Salmonella enterica* [28], etc.; as well as microorganisms associated with food spoilage: *Pseudomonas*, *Acinetobacter*, *Flavobacterium*, *Enterococcus*, etc. [29,30]. The BFs studied in this work contained representatives of 11 phyla.

It should be noted that food BF formation is strongly influenced by components of food residues, including meat residues and meat exudates [31]. It was confirmed in this work. Samples taken in the deboning and trimming workshop (on the cutting table and on the surface of the drain), which had the greatest diversity of the biofilm population (Table 2), contained meat cells included in the biofilm matrix (Figure 1 and Figure 3). At a higher magnification, membrane vesicles were found on the cells, as well as in the matrix. Vesicles found in BF are considered to be derivatives of the outer membrane of gram-negative bacteria, which are associated with the transport of exoenzymes and endotoxins from the cell [32,33].

The literature indicates that the nature of the food matrix affects the selective growth and biofilm formation by certain bacteria species (genera) [34]. Thus, some representatives of the *Pseudomonas* genus (*Proteobacteria* phylum) cause spoilage of meat products [35]. Comparative analysis of BF metagenomes (Figure 7) sampled in different places of the meat processing plant (Table 1) confirms this pattern. In all studied biofilms, representatives of *Bacteroidetes* (22.3–67.7%), *Proteobacteria* (8.5–42.8%), *Firmicutes* (0.6–32.8%), *Actinobacteria* (1.8–36.3%) were dominant. Our microscopic analysis of BFs from different samples also indicated the selective dominance of certain bacteria according to their morphological characteristics (Figure 1, Figure 2, Figure 3, Figure 4, Figure 5 and Figure 6).

Representatives of the *Pseudomonadaceae* were found in different quantities in all the samples studied. Representatives of *Pseudomonadaceae* are aerobic psychrophilic eurybionts. Some representatives of this genus cause spoilage of dairy [36], fish [37], and meat products [38]. The presence of bacteria of the *Pseudomonadaceae* in all samples taken at the meat-processing plant is of concern due to the negative impact that it can have on the quality of food, having a high ability to form biofilm, as well as cause spoilage of food [39]. *Proteobacteria*, which include the *Pseudomonadaceae*, actively grow in cold conditions and are able to form biofilms on surfaces directly contacting with food [39], and to tolerate some disinfectants, especially quaternary ammonium compounds [40], which are broadly used during the sanitation process in the food industry [41].

*Pseudomonas* spp. produce vast amounts of extracellular polymeric substances (EPS) and are known to most commonly adhere to and form biofilms on stainless steel surfaces. They can coexist with other pathogens in biofilms, forming multi-species biofilms, making them more resistant and stable [42].

Our results are consistent with previous studies in which this bacterium was found to be the most predominant in biofilm samples on surfaces in contact with food [43]. In the study of biofilms selected at four food processing enterprises, two genera, *Pseudomonas* spp. and *Acinetobacter* spp., were most frequently detected (93.47%) [44]. Some authors report the dominance of *Pseudomonas* spp. over other bacteria during biofilm formation in vitro [45].

The *Flavobacteriaceae* family also was represented in all samples (phylum *Bacteroidetes*). Representatives of this genus are widespread in water. It is known that some *Flavobacteria* lead to spoilage of fish products, however, their activity does not cause significant harm to food industry [46]. Various *Flavobacterium* strains are able to form biofilms [47]. It has also been shown that *Flavobacteria* may deteriorate canned milk [48].

The dominant species in sample 2 were representatives of the family *Crocinitomicaceae* (genus *Fluviicola*), which are often present in wate [49] and soil [50] samples. *Fluviicola* are morphologically similar to *Flavobacterium* [51]. It should be noted that representatives of this genus were found in wastewater biofilms [52], however, their role in biofilm formation is not clear.

The dominant families in sample 4 were representatives of the following phylum: Actinobacteria-Corynebacteriaceae (16.29%), Actinobacteria-Micrococcaceae (15.04%), Firmicutes-Enterococcaceae (12.47%), Bacteroidetes-Sphingobacteriaceae (11.11%), Bacteroidetes-Weeksellaceae (10.36%).

There are human pathogens among bacteria of the *Corynebacterium* genus. Representatives of the *Chryseobacterium* genus have been found in various ecotopes including foods [43]. It has been shown that activity of *Chryseobacterium joostei*, along with *Pseudomonas fluorescens*, causes spoilage of milk [53], as well as contamination of animal and poultry meat products, when entering from the environment. Representatives of the *Carnobacterium* genus, which dominate in 5 sample, are also typical for meat, fish and dairy products and cause their spoilage [54].

*Enterococcaceae* were also represented in samples 2 (4.90%), 3 (1.09%) and 4 (12.47%), and in a minimal amount in sample 1 (0.13%). Members of the family such as the genus *Enterococcus* and *Vagococcus* are able to form a biofilm on different surfaces [55,56]. It is assumed that *Vagococci* may be involved in the spoilage of meat and fish products. Thus, *Vagococcus penaei* strain was isolated from spoiled shrimp [57]. Bacteria of the *Vagococcus* genus were also found in microbiota samples from fresh broiler carcasses [58].

Probably, the high diversity of the community determines the formation of biofilm with chemically heterogeneous matrix, which protects biofilm bacteria from external environmental factors, but does not allow active development of minor representatives. It is worth noting that a number of minor microorganisms in the considered communities are also bacteria typical for food biofilms, for example, families *Aerococcaceae* and *Carnobacteriaceae* were found at all sampling points except for sample 1. Representatives of the *Clostridiaceae* were found in samples 2, 4 and 5. The largest number of them was presented in sample 5 and amounted to 1.96%. Representatives of the family *Lactobacillaceae*, *Listeriaceae* and *Staphylococcaceae* were found mainly only in sample 4, which was selected in the forming workshop at the corner of the vacuum filler table.

Lactic acid bacteria (LAB) are a group of microorganisms capable of growing at low temperatures, and some species are relatively tolerant to stressful conditions [59], which provides them with a high ability to spoil food [60].

*Enterobacteriaceae* is a bacterial family of great importance in the food industry, as they are indicators of improper hygiene. Many members of the family, such as *Salmonella* spp. and *E. coli* O157:H7, are considered foodborne pathogens, and some species are recognized as spoilage microorganisms [60]. The main genera of the *Enterobacteriaceae* family found in this study were *Serratia*, *Morganella* and *Budvicia*. In accordance with our results, some authors also reported a high prevalence of representatives of the genus *Escherichia* in the food industry [61]. High prevalence of *Serratia* spp. It is consistent with what has been reported in other studies where this genus has been more identified at various food processing enterprises [62].

Despite the fact that in our study food-borne pathogens such as *L. moncytogenes*, *Salmonella* spp., *S. aureus* and *Campylobacter* spp. were not detected in biofilms, we cannot ignore the protective and synergistic effect that biofilms offer for these microorganisms, which poses a constant risk to safety and the quality of processed foods.

The next characteristic of the sampled food BFs is the microcolonial growth of biofilm bacteria, which is clearly seen in Figure 3, Figure 4, Figure 5 and Figure 6. At the same time, cells of different physiological ages coexist within the same microcolony: (a) vegetative cells with a structure typical mainly for gram-negative bacteria, with well-distinguishable external and cytoplasmic membranes and a dispersed nucleoid (Figure 3, Figure 4, Figure 5 and Figure 6); and (b) resting forms characterized by thick membranes, electron-dense cytoplasm and compacted nucleoid located in the cell center (Figure 3, Figure 4, Figure 5 and Figure 6). Biofilms of sample 1 contained cells covered with a well-detectable fibrillar layer stained with ruthenium red. Such “hairy” cells covered with a dense fibrillar layer (FL) were first discovered by us as a characteristic morphotype of type I persister cells of *S. aureus* [63]. Persisters were found in a culture grown on rich medium (LB) after exposure of 24-h culture (stationary phase) to ciprofloxacin as an antibacterial selective agent that kills ordinary vegetative cells and keeps antibiotic-resistant persister cells in native form [9,10]. However, in *S. aureus* persisters, this fibrillar layer was much denser. Another analogy is *Salmonella* cells with a fibrillar layer, possibly also of the persister phenotype, found in monospecies *Salmonella* biofilms developing on plastic and in planktonic culture [16]. Persisters of gram-negative bacteria *E. coli* [64] and *P. aeruginosa* [65] did not have such a fibrillar layer. It may be assumed that the “hairy” cells found in the biofilm of sample 1 (Figure 1 and Figure 2) belong to a gram-positive bacterium, which does not contradict the metagenomic analysis of this sample (Appendix A).

According to the extensive literature [3], the revealed heterogeneity of food biofilm bacteria, both in taxonomic classification and physiological age, is important in the resistance of the biofilm consortium to damaging effects and disinfectants. Despite the large amount of information obtained, the mechanisms of BF resistance are still unclear. The most proven explanation for the high resistance of the biofilm population to disinfectants and antibiotics is the development of polymer matrix, which prevents toxicant contact with bacteria. Within this explanation, it is assumed that in multi species BFs, the chemical nature and physicochemical properties of matrix biopolymers vary greatly compared to the composition and properties of mono species BF matrix. It is believed that matrix properties vary depending on the environmental conditions and bacteria type [2], while in multispecies BFs, chemical and physicochemical interactions between matrix polymers synthesized by different bacteria may cause high matrix viscosity providing better protection of BF consortium cells [66]. In food multispecies BFs analyzed by us, a similar heterogeneity of the matrix was noted evidenced by its staining with ruthenium red (Figure 3, Figure 4, Figure 5 and Figure 6).

A further explanation for the high resistance of multispecies BFs is that the organization of microorganisms in BF is not accidental. Bacterial cells of the same species are organized in microcolonies coexisting side by side. In this work, it was demonstrated, for example, in Figure 5, where bacteria of 5 morphotypes are present. In addition, the microcolonial growth of biofilm bacteria is demonstrated in Figure 6, which shows cross sections of compactly packed microcolonies.

However, recently, more studies indicate that multispecies BFs are not always more resistant to damaging influences than monospecies BFs, and that resistance depends on: (1) bacteria species; (2) their physiological age (dividing or stationary) [67,68], and (3) their phenotype. Thus, persister cells and resting forms developing from them found in BFs differ in tolerance to antibiotics and disinfectants [69,70]. Age heterogeneity in multispecies food BFs was demonstrated by us for all studied samples (Figure 1, Figure 2, Figure 3, Figure 4, Figure 5 and Figure 6).

It should be emphasized that the growth of microorganisms, including food pathogens, in the form of multispecies BFs contributes to the development and spread of resistance to toxicants due to both the horizontal exchange with genetic determinants of resistance (facilitated by the high density of microbial population in BFs) and the development of persisters preserving and passing new determinants of resistance to next generations [71,72]. This creates new foci of biofilm formation and cross-contamination. Thus, there is the need for: (a) correcting sanitary procedures, (b) developing alternative strategies to improve the effectiveness of biofilm formation prevention [68], and (c) developing methods and tools to minimize the formation of persister cells in BFs as the main phenotype of genetic resistance development.

## 5. Conclusions

In conclusion, this study confirmed the frequent presence of multi-species biofilms in the meat processing environment. With the help of electron microscopy, the main features of their structure were characterized:(1)diversity of cell morphotypes in multi-species BFS;(2)certain morphological similarity of cells in BFS from different samples and microcolonial growth;(3)age heterogeneity of cells within a single microcolony (vegetative and autolysed cells, resting forms);(4)heterogeneity of the chemical nature of the polymer matrix (polysaccharide or non-polysaccharide nature,

Understanding the development and composition of biofilms in the food industry environment will help us prevent contamination by preventing the formation of biofilms and removing biofilms.

## Figures and Tables

**Figure 1 microorganisms-10-01583-f001:**
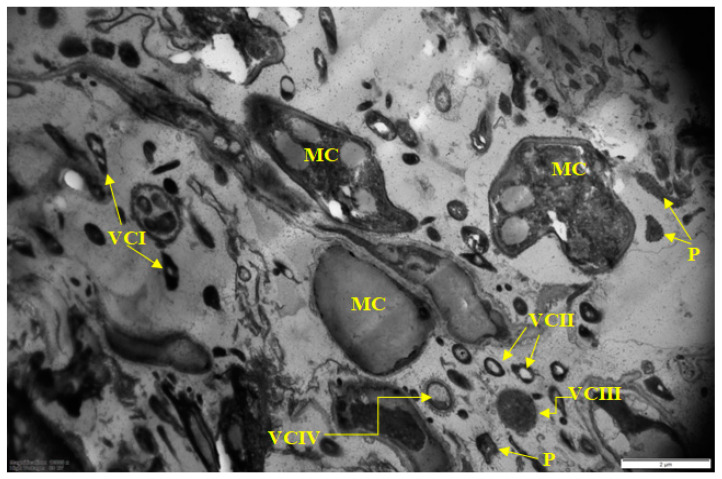
Electron micrograph of sample 1 biofilms. Designations: VC I; VC II; VC III, VC IV—vegetative cells of I–IV morphotypes, respectively; P—persisters covered with a fibrillar layer; FL—fibrillar layer; MC—meat cells. Scale bar is 2 μm.

**Figure 2 microorganisms-10-01583-f002:**
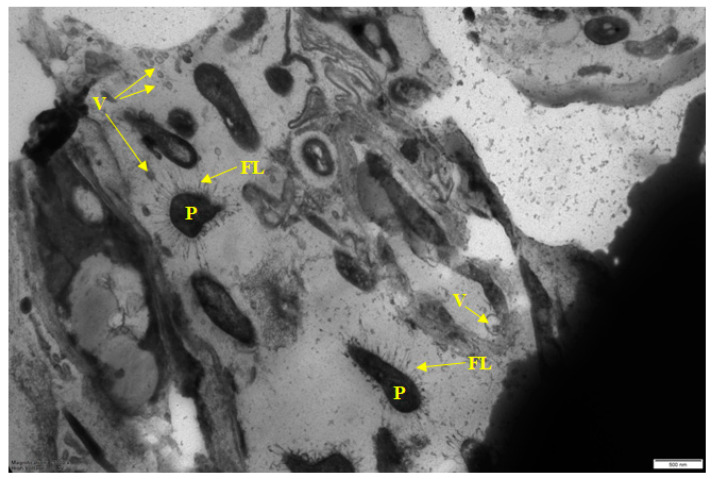
Electron micrograph of sample 1 biofilm (magnified). Designations: P—persisters covered with a fibrillar layer; FL—fibrillar layer; V—vesicle. Scale bar is 0.5 μm.

**Figure 3 microorganisms-10-01583-f003:**
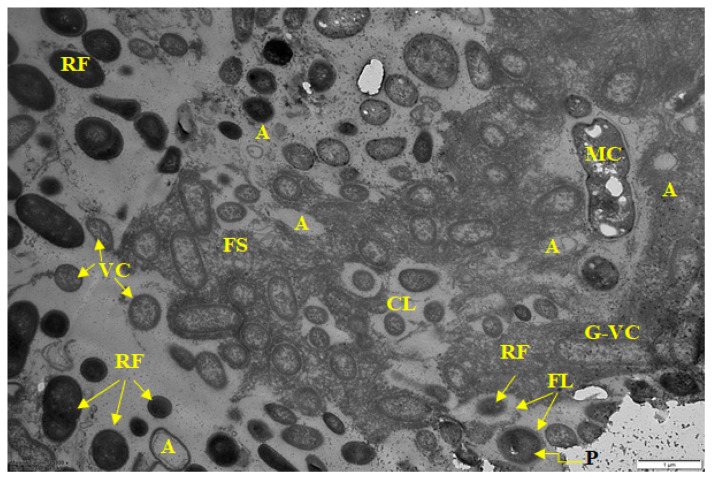
Electron micrograph of sample 2 biofilm. Designations: VC—vegetative cells; G-VC—vegetative cells of gram-negative bacteria; RF—resting forms; FS—fibrous structures of the matrix; CL—capsular layer; P—persisters; FL—fibrillar layer; A—autolyzed cell. Scale bar is 1 μm.

**Figure 4 microorganisms-10-01583-f004:**
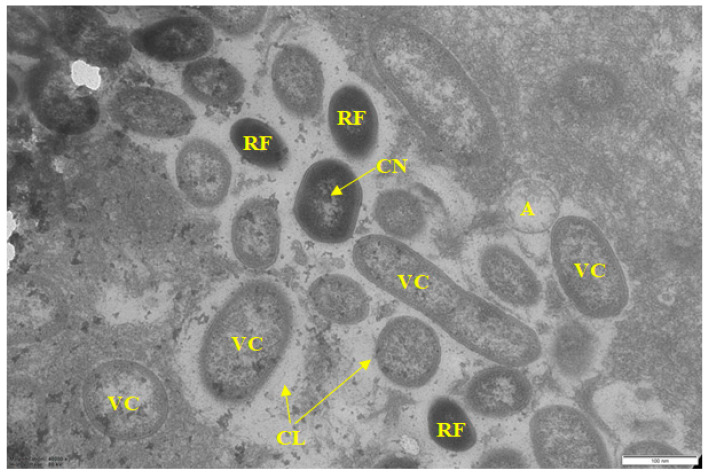
Electron micrograph of sample 2 biofilm (magnified). Designations: VC—vegetative cells; RF—resting forms; CL—capsular layer; A—autolyzed cell: CN—compacted nucleoid. Scale bar is 0.5 μm.

**Figure 5 microorganisms-10-01583-f005:**
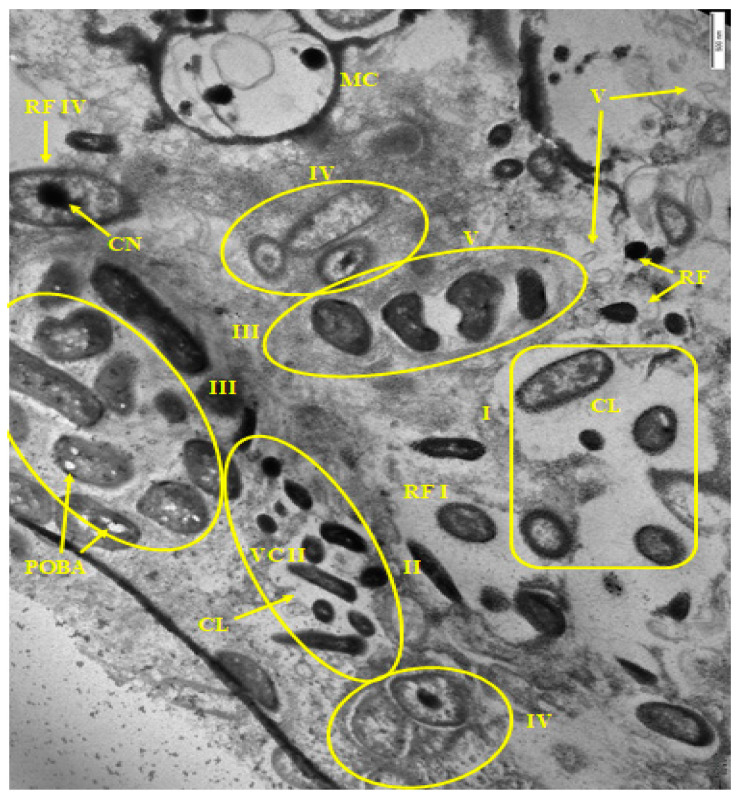
Electron micrograph of sample 4 biofilm. Designations: VC—vegetative cells; RF—resting forms; CL—capsular layer; I, II, III, IV, V—microcolonies of I–V morphotype cells, respectively; V—vesicles; POBA—polyoxybutyric acid, MC—meat cell. Scale bar is 0.5 μm.

**Figure 6 microorganisms-10-01583-f006:**
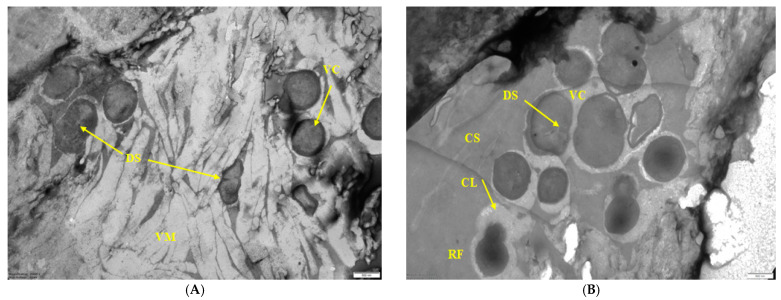
Electron micrographs of sample 5. (**A**)—zone of BF with veiny material; (**B**)—zone rich of cells in carbohydrate matrix. Designations: M—matrix (veiny structures VM or carbohydrate structures CS); VC—vegetative cells, and RF—resting forms of the same microcolony; DS—division septum; CL—capsular layer. Scale bar is 0.5 μm.

**Figure 7 microorganisms-10-01583-f007:**
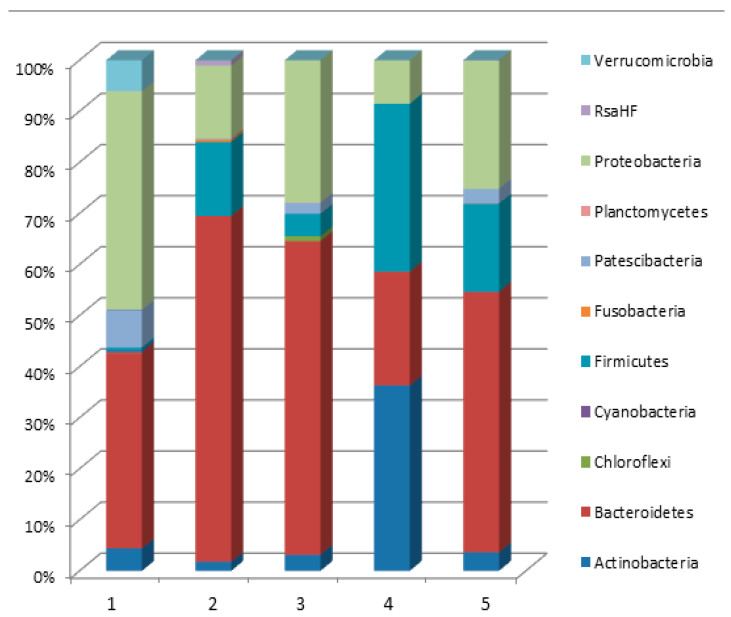
Taxonomic composition of microbial community of biofilms.

**Figure 8 microorganisms-10-01583-f008:**
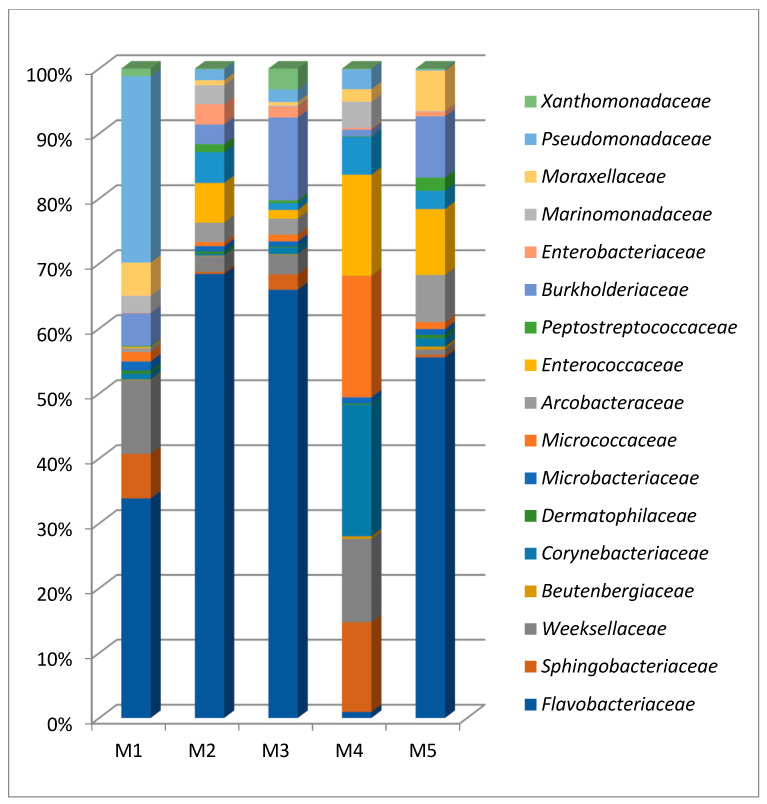
The general of the families found in each samples according to relative abundance. M1, M2, M3, M4 and M5 represent each food processing specific location (*x*-axis). Color boxes represent bacterial abundance (*y*-axis).

**Table 1 microorganisms-10-01583-t001:** Biofilms sampled for research at meat processing plant.

No. of Sample	Workshop	Specific Location	Appearance
1	Deboning and trimming workshop	The joint of the wall and the cutting table	Dense, thick
2	Deboning and trimming workshop	Drain	Wet, thick
3	Deboning and trimming workshop	The joint between tiles	Dry
4	Forming workshop	Equipment—vacuum filler, table corner	Wet, thin
5	Forming workshop	Equipment—vacuum filler, inner surface	Leathery, hard

**Table 2 microorganisms-10-01583-t002:** Indices of species diversity for biofilms #1–#5 (correspond to biofilms described above).

Sample	Chao1	Richness	Richness2	Shannon_e
1	324.5	231	164	3.99
2	169.2	140	101	3.43
3	219.3	197	161	3.13
4	123.4	93	66	2.92
5	192.3	152	115	2.98

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
