# Peer review of "Microbial Biofilms at Meat-Processing Plant as Possible Places of Bacteria Survival"

_microorganisms, 2022, doi:10.3390/microorganisms10081583_

Round 1
Reviewer 1 Report
The present manuscript concerns the main findings of multi-species biofilms through electron microscopic analyzes of different samples collected at a meat processing site, as well as metageomic analyzes for taxonomic identification of the samples.
I indicate acceptance of the manuscript after minor revision considering the following points:
- A small revision in the writing of the text should be carried out to repair lack of punctuation (eg lines 168, 272), inappropriate use of highlighting, space or dash (eg lines 262, 291, 302, 402, 425). I strongly suggest the revision for correct writing and abbreviation of phylum, family, genus and species of microorganisms, respecting the rules of use of italics and abbreviation (for example lines 286, 287, 290-295, 297, 298, 315-317, 322 , 324, among others). Considering that this is a taxonomy study, the mastery of correct writing and standardization of it is essential.
- Table 2 can be transferred to supplementary materials, as chemicals are mentioned in the text;
- According to the authors, the samples were taken before the routine disinfection procedure (lines 90-91) and the studied MP complied with a strict with sanitization and disinfection measure (lines 327-328). Moreover, the revealed heterogeneity of food biofilm bacteria, both in taxonomic classification and physiological age, is important in the resistance of the biofilm consortium to damaging effects and disinfectants (465-467). Considering these citations and the fact that biofilms persist after the use of disinfectants, a paragraph should be included in the text explaining whether biofilms were observed after the disinfection procedure and why the same microscopic and metagenomic analyzes were not performed after the use of disinfectants.
Author Response
Comment 1: Corrections were made.
Comment 2: We thank your for this comment. Corrections were made.
Comment 3: Table 2 was transferred to supplementary materials.
Comment 4: We thank the reviewer for this valuable comment. Appropriate paragraph has been inserted in the beginning of 2.1 section. Indeed, it is a good idea to investigate BF’s structure and composition in dynamics after disinfection in addition to work done. We’ll do it in another separate experiment. Current experimental design was chosen because of known high resistance of BF to disinfectants, and our research confirmed this too (in another paper we’ll describe these data). We also draw attention of respectable Reviewer #1 to the fact of old mature BF presence (as indicated by polyoxybutyric acid and resting forms presence). This is in agreement with our assumption that disinfection doesn’t kill BFs.
Reviewer 2 Report
The article by Nikolaev et al. and entitled “Taxonomic composition and structural characteristics of microbial biofilms at meat-processing plant as possible places of pathogenic bacteria persistence” has as main aim to detect biofilms in various technological zones of a meat processing plants as a possible source of contaminating microflora and to determine biofilm consortium taxonomic composition, structural organization, morphological and age heterogeneity. However, along the manuscript nothing is described as a detection method of Food biofilms. Moreover, authors collected samples before the routine disinfection procedure (daily treatment of all technological surfaces with sodium hypochlorite solution) from technological equipment or premise surfaces. Thus, in the routine of the companies, those samples will not contaminated food because after those biofilms were formed they will be subjected to disinfection! It would be of much more interest to try to collect biofilms after disinfection procedure because that is the real situation in food facilities.
But, in discussion section (lines 328-39), authors said that “It should be noted that despite strict compliance with sanitization and disinfection measure at the studied MP, BFs were detected as early as 10 to 24 hours after sanitization”. So, I don't understand what they did.
Please clarify this.
Overall, the manuscript is well written, describes an interesting subject and presents interesting findings.
Main Comments:
I think that the Title is somehow confuse. Please reformulate it.
Introduction: This work is about the problem of contamination of food with pathogenic bacteria from surfaces, i.e., is about the problem of cross-contamination in food industry. As Cross- contamination is only referred to in line 498, in the discussion, please include some text about it in the Introduction.
Line 25 – Replace “sec-tions” by “sections”
Line 42-43: The sentence “Biofilm contamination, including pathogenic bacteria, at food production facilities is a constant source of microbial contamination” makes no sense.
Line 55 – Please explain why antibiotics are biological toxicants and disinfectants are nonbiological toxicants. As far as I know antibiotics are not produced by microorganisms. Please remove this classification from all the text or clarify it.
Table 2: Replace OsO4 by Osmium oxide
Line 115-116: Replace the sentence…”during one day and then at 60 °C during one day” by “ …. during one day and then at 60 °C for another day”
Authors classified some detected cells in biofilms as persister cells . How can they say that those cells are persister? As far as I know, persisters are thought to originate from dormant cells in which antibiotic targets are less active and cannot be corrupted. Bacterial persistence refers to the capacity of small subpopulations within clonal populations to tolerate antibiotics. But in this study it was not studied the tolerance to antibiotics, so….
However, in discussion it is said that “Persisters were found in a culture grown on rich medium (LB) after exposure of 24-hour culture (stationary phase) to ciprofloxacin as an antibacterial selective agent that kills ordinary vegetative cells and keeps antibiotic-resistant persister cells in native form”. This information must be included in Materials and Methods section. In fact, when authors identified some cells in the figures as persisters I already asked to myself how did they know. Thus, they have to described in materials and methods section how to identified persisters and the other type of cells.
Line 311 - Replace “do- mi-nant” by dominant
Line 397 – Replace “wate” by “water”
Line 439: Remove “It” after Serratia spp.
Line 442 – L. monocytogenes instead of L. moncytogenes and, please, in italic.
Line 442 – S. aureus instead St. aureus and, please, in italic.
Author Response
Comment 1: We agree and the Title was reformulated.
Comment 2: Some text included into the Introduction.
Comment 3: We corrected the Manuscript accordingly.
Comment 4: We corrected the Manuscript accordingly – confusing classification was removed.
Comment 5: We corrected the Manuscript accordingly.
Comment 6: We thank the respectable reviewer for this multi-aspected comment. Our answer is the following:
*… We consider observed “hairy” cells to be persisters on the base of our previous investigation of Staphylococcus aureus persister cells [63]. Indeed, it is only our presumption, not the proof. We believe that in this research paper it is appropriate to discuss such things. We wrote a new paragraph in 2.3 section.
**…Indeed, at this stage of our research we did NOT prove the nature of hairy cells, because of the aim of the work was another. In our current investigation, we strictly proven the presence of persisters in BFs with the use of antibiotics. I’m sure soon you’ll be able to read and evaluate it.
***.. The cited sentence describes literature data, so it was not included in Materials and Methods section.
****. We wrote a new paragraph in 2.3 section.
Minor but nerving mistyping were also corrected.

Reviewer 3 Report
Biofilm in the food industry is a problem. Nevertheless, this experimental design does not bring any novelty in the process of how to combat it. The NGS method should use a longer part of 16S rDNA (different primers). It is almost impossible to find pathogens by this method. The methods should be described more clearly. The used references should be more recent and relevant to the research - should be extensively reduced. Taxonomic names of microbes are not well-written - they should be in italics and with the correct spelling and uniform throughout the manuscript. TEM does not belong to light microscopy (3.1 - first paragraph).
These results are of low importance in bringing new knowledge.
Author Response
Thanks for your comments. We will try to take them into account in the next work. We would like to clarify the following:
- And 7. To prove absence of novelty and “low importance in bringing new knowledge” one should cite works where structure of native BF of meat plant(s) and persisters and resting forms in BF and other observations were described. Combatting BFs was not an aim of this work, though any knowledge will help to fight them.
- We absolutely agree with the comment. The method based on the V3-V4 variable fragment used in the NGS work provides information exclusively on the taxonomic composition of the microbial community. It is impossible to find pathogenic organisms using this approach. In our work, we used this approach exclusively to describe the taxonomic composition of the studied samples and to suspect pathogens presence. Comparison of these data with the results of microbiological and microscopic studies provides the most complete information about the properties of the studied community, which is what we did in the work described.
- This point is not clear. What is wrong is not specified.
- This is not clear either. It does’ not contradict Journal’s rules. Irrelevance to the research must be proven by some examples.
- Corrections were made.
- Mistyping was corrected.
Round 2
Reviewer 2 Report
After carefully reading the revised manuscript, I believe that the authors have improved it and have taken into account all the suggestions I made.
Author Response
Thank you for reviewing and recommending the publication of our article.
Reviewer 3 Report
P446 - ssp. should ve written without italics asi it means species in plural. Please, do not write St. aureus, but S. aureus, as it Is right according the taxonomy - corret in whole text.
Author Response
Thank you for reviewing, we took into account your comments and made changes to the text of the latest edition.